# Comprehensive Performance Analysis of Zigbee Communication: An Experimental Approach with XBee S2C Module

**DOI:** 10.3390/s22093245

**Published:** 2022-04-23

**Authors:** Khandaker Foysal Haque, Ahmed Abdelgawad, Kumar Yelamarthi

**Affiliations:** 1Institute for the Wireless Internet of Things, Northeastern University, Boston, MA 02115, USA; haque.k@northeastern.edu; 2College of Science and Engineering, Central Michigan University, Mt Pleasant, MI 48859, USA; abdel1a@cmich.edu; 3College of Engineering, Tennessee Tech University, Cookeville, TN 38501, USA

**Keywords:** Zigbee, testbed, experimental, wireless sensor network, power consumption, RSSI, latency, throughput

## Abstract

The recent development of wireless communications has prompted many diversified applications in both industrial and medical sectors. Zigbee is a short-range wireless communication standard that is based on IEEE 802.15.4 and is vastly used in both indoor and outdoor applications. Its performance depends on networking parameters, such as baud rates, transmission power, data encryption, hopping, deployment environment, and transmission distances. For optimized network deployment, an extensive performance analysis is necessary. This would facilitate a clear understanding of the trade-offs of the network performance metrics, such as the packet delivery ratio (PDR), power consumption, network life, link quality, latency, and throughput. This work presents an extensive performance analysis of both the encrypted and unencrypted Zigbee with the stated metrics in a real-world testbed, deployed in both indoor and outdoor scenarios. The major contributions of this work include (i) evaluating the most optimized transmission power level of Zigbee, considering packet delivery ratio and network lifetime; (ii) formulating an algorithm to find the network lifetime from the measured current consumption of packet transmission; and (iii) identifying and quantizing the trade-offs of the multi-hop communication and data encryption with latency, transmission range, and throughput.

## 1. Introduction

Internet of Things (IoT) is a network of interconnected objects where these objects/things transfer data among themselves. The recent development of wireless communication has prompted the IoT to grow multi-fold in the last few years. It has diversified application fields, including smart cities, home automation, industrial manufacturing, intelligent transportation systems (ITS), vehicular communication, and smart agriculture [1,2,3]. In all of these applications, wireless sensor networks (WSNs) are deployed to interact with the surroundings and perform the assigned tasks. WSNs consist of sensor nodes that communicate wirelessly to transfer data, are deployed in remote locations, and are battery powered. Thus, they have constrained power supplies and must operate on low power modes. Zigbee communication protocols is one popular standard used for such a low-powered wireless communication, based on IEEE 802.15.4 [4,5]. ZigBee is currently maintained by Zigbee Alliance and built on top of the IEEE 802.15.4 physical (PHY) layer and medium access control (MAC) layer, as depicted by Figure 1 [6]. The PHY and MAC layer defines the low-level network procedures, such as transmission, selection of channels, and operating frequency. The network (NWK) layer and the application (APL) layer are defined by the Zigbee specifications. The NWK layer defines the architecture of the network, routing protocols, and security and, thus, encryption and decryption of the transmission. The APL layer has three sub-layers, namely application support, application framework, and the Zigbee device object. The application support sublayer interfaces the NWK layer with the APL layer whereas the application framework forms the environment to host the application objects. On the other hand, the Zigbee device object sublayer allows advanced networking along with device and service discovery features [7].

Zigbee is a short-range communication standard based on personal area network (PAN) and operates on the industrial, scientific, and medical (ISM) band of 2.4 GHz [8]. It also operates on 868 and 915 MHz frequency bands with the data rate ranging from 20 to 250 Kbps. It operates on 27 different channels, of which 16 are in 2.4 GHz, 10 channels are in 915 MHz, and 1 channel is in the 868 MHz band [9,10]. The networking characteristics of Zigbee are depicted in Table 1. Zigbee is used in different interesting real-world applications, such as seismic data acquisition, the mining ventilation system, radio frequency fingerprinting, and different biomedical applications [11,12,13,14,15]. Due to its interoperable standards and low power operation modes, Zigbee has been popular in indoor, industrial, and outdoor applications.

Thus, it has drawn the attention of both industry and academia. Extensive research is being conducted to improve the applicability, network optimization, and performance of Zigbee. The performance of Zigbee depends on various parameters, such as transmission power, deploying scenario, transmission distance, data encryption, baud rates, and the number of hops for the communications. Variations on these parameters greatly affects the performance metrics, such as latency, throughput, link quality, and power consumption. For optimal designing of the network, a clear understanding of the behaviors of these preference metrics corresponding to the above-mentioned parameters is necessary. While research has been conducted to identify the performance analysis of Zigbee, the existing research does not present a comprehensive evaluation for both indoor and outdoor environments while considering all the performance metrics. Encryption is another important factor that provides secured communication with Zigbee but limits the performance of the Zigbee in terms of latency, throughput, and power consumption. It also has not been addressed yet in the performance analysis, which is necessary for better understating of the trade-offs of the encrypted communication corresponding to the performance metrics.

Thus, we conducted an extensive performance analysis of both the encrypted and unencrypted Zigbee communication in both indoor and outdoor environments while accounting for performance metrics, such as PDR, network lifetime, latency, throughput, link quality, and power consumption, by varying the networking parameters, such as baud rates, transmission distance, transmission power, experimental scenario, and hopping (single and multi-hop) of the communication. The primary contributions of the work include:We analyzed the PDR, energy consumption, and network lifetime for the different transmission power levels of the XBee S2C module; evaluated the optimized power level based on the performance and trade-offs.We developed an algorithm to measure the node lifetime and we verified the current consumption through an experimental testbed.We analyzed link quality in terms of the received signal strength indicator (RSSI) for both indoor and outdoor environments with different transmission power levels and the number of hops. This presents a detailed study of how the tx power, network environment, and hopping impact the link quality.Latency was analyzed for different baud rates and packet sizes in both indoor and outdoor environments with encrypted and unencrypted communication. This depicts the trade-offs among latency, encryption, multi-hopping, and packet sizes.Throughput evaluation was performed via an experimental testbed at various baud rates to identify the trade-offs between packet size, encryption, and throughput at various indoor and outdoor scenarios.

The rest of the paper is organized as follows: the background and related works are presented in Section 2; the experimental setup and procedures are depicted in Section 3; Section 4 presents the performance evaluation; and the paper is concluded in Section 5.

## 2. Background and Related Works

### 2.1. Background on Zigbee Communication

A Zigbee node in a network can be configured with three different modes: (i) coordinator, (ii) router, (iii) end nodes [17]. The coordinator is the central node of the network that acts as the gateway, and it is responsible for allowing any node to join the network. Data can be broadcast from the coordinator to every node of the network and information can be sent to any node from the coordinator. It can also change the configuration of any of the member nodes with over the air (OTA) programming. The function of the router nodes is to route the data from the coordinator to the designated end node or vice-versa. We should note that, they can also be equipped with sensors to sense and transmit the data to the coordinators. However, in this study, the routers were not equipped with any sensors to keep the number of packets and transmitting bytes the same while conducting the measurements with or without router nodes (single and multi-hop). This was done to make the comparison fairer with single and multi-hop communication. End nodes are edge nodes of the WSN and are usually equipped with sensors to interact with the surroundings. End nodes can be configured to deep sleep, cyclic sleep, and reserved mode to conserve the battery power [18]. Zigbee standard allowed for communication in three different topologies: (i) star, (ii) tree, and (iii) Zigbee mesh topology [19], as in Figure 2. A Zigbee star network consists of one coordinator and a few end nodes connected directly to it. A Zigbee mesh network is self-healing by nature and consists of one coordinator and multiple routers and end nodes. Routers can communicate among themselves and with the end nodes to route the data from the end node to the coordinator or vice-versa.

Security is a critical measure in wireless communication and Zigbee facilitates secured communication with 128-bit key advanced encryption standard (AES) [20,21]. The AES algorithm is used for encrypting the data and also for checking the data integrity with a message authentication code, which is formed by encrypting the IEEE MAC frame. The AES-CTR mode is used in encrypting the payload. Zigbee facilitates two additional security layers in the NWK and APL layers in comparison to the IEEE 802.15.4, which are also based on AES 128-bit encryption. Though these encryptions provide much needed security, they limit the latency, throughput, and longevity of the network.

Zigbee is a low-power PAN protocol and has great potential in both indoor and outdoor applications [22,23,24,25]. The performance of the indoor deployment varies from that of outdoor deployment due to the presence of obstacles and interference of other radio frequency carriers. The indoor applications of Zigbee include home automation, smart surveillance, energy management, indoor localization, and many more where the end nodes and the router are most likely to be placed with a non-line of sight (NLoS) medium due to the infrastructure [26,27,28,29]. Firstly, this NLoS medium affects the link quality, transmission range, and throughput due to the multipath propagation and hindrance provided by the walls or any other metallic infrastructures. Secondly, as the most widely used operating frequency of Zigbee is 2.4 GHz, which is the same as Wi-Fi, Bluetooth, and Z-Wave, this would create cross technology interference (CTI), which might drastically affect the Zigbee performance [30,31].

Baud rate is another important parameter that can influence the performance parameters significantly. In particular, the latency and throughput greatly depend on it. The bit rate of the transmission greatly depends on the baud rate and there is a very close relationship between them [19]. The baud rate is the transmission rate of the signal unit or symbol, whereas the bit rate is the transmission rate of the bits. This can be expressed with Equation (1), as follows:(1)$$\mathrm{Bit}\;\mathrm{rate}=\mathrm{Baud}\;\mathrm{rate}\times \frac{\mathrm{Bits}}{\mathrm{Symbol}}$$

Zigbee operates with different baud rates ranging from 4800 to 115,200. Moreover, 9600 and 115,200 are two widely used baud rates as most of the associated sensors work with these two. Thus, this study considers different baud rates along with transmission power level, transmission distance, surroundings (indoor/outdoor), and data packet size for performance evaluation of the Zigbee Quality of Service (QoS).

### 2.2. Related Works

Substantial research is being conducted with the performance analysis of Zigbee under various constraints, scenarios, and applications. Hamdy et al. have studied the throughput and latency of the Zigbee WSN under three different topologies and concluded that at 2.4 GHz, operating frequency tree topology achieves the highest throughput whereas the star topology performs with the least latency [17]. However, this study is based just on simulation, and is only focused on the performance of the latency and throughput. Desnanjaya et al. conducted a study on the transmission range of the Zigbee communication with the XBee pro 2B module for both indoor and outdoor scenarios [32]. However, they did not consider other performance metrics beyond the transmission range. Fitriawan et al. conducted a performance analysis of Zigbee in various indoor setups to study different network Quality of Service (QoS)-delays, throughput, and packet loss [19].

Even though this study considered LoS and NLoS scenarios for the analysis, the performance with encryption and outdoor scenario was not addressed. Varghese et al. conducted a comparative analysis on the Zigbee QoS with three different topologies in an IoT lighting automation network [27], studied the performance of two different routing protocols—ad-hoc on-demand distance vector (AODV) and dynamic source routing (DSR)—and evaluated that star topology with DSR routing performs better in end-to-end delay, throughput, and latency with the trade-off of higher energy consumption. However, this study is based just on a simulation. Moulik et al. studied the superframe structure of the MAC sublayer of Zigbee and made a point that the performance of such a low-rate PAN network largely depends on the active portion of the superframe [33]. Soijoyo et al. conducted a study on different Zigbee topologies in the LoS scenario based on the performance in latency, throughput, and packet loss [34]. The study concluded that star topology performs better in short-distance communication, but tree and mesh topology would be necessary for larger network coverage. However, this study did not consider the parameters, such as RSSI and energy consumption, and the study was not compared with the NLoS scenario with data encryption, which would have provided a better understanding of the QoS performances. Khalifeh et al. conducted a study on the performance of two Zigbee-based mesh topologies, Zigbee mesh and Digi mesh, and concluded that Zigbee mesh performs better with time-critical applications due to lower latency and better link quality, but better throughput is achieved with Digi mesh [35]. This study is limited by the lack of different deployment scenarios, indoors and outdoors, where the performance might have varied significantly. Moreover, it did not study the power consumption and performance of encrypted communication, which is needed to reduce the security threat. Moridi et al. analyzed different Zigbee topologies with underground deployment by considering throughput, delay, energy consumption, delivery ratio, and packet delivery security [32]. This analysis is based on simulations whereas the real-world performance might vary from the simulated results. Mounika worked on a simulation-based performance analysis with Zigbee topologies using the Riverbed simulation [36]. This work analyzed how the network performance changes with the increase in the number of nodes in three different topologies, star, tree, and mesh, and their results showed that the throughput of the mesh topology was higher than that of the tree and star topologies. Rao et al. also used the Riverbed simulator to analyze the performance of different topologies of Zigbee and concluded that cluster tree performs better in scenarios where the larger network coverage is needed [37]. Few other research works on the Zigbee performance analysis are stated in [38,39,40]. The discussed research works are summarized in Table 2.

Most of these performance analyses are based on simulations whereas the real-world performance with the deployed network might significantly vary from the simulated results. Some of the research works are conducted in the indoor environment but those did not analyze how the performance might vary in the outdoor scenarios. None of these discussed research works analyzed the performance effects of encrypted communication even though it may significantly hamper the throughput, latency, and network lifetime. Thus, this research work provides a comprehensive analysis of Zigbee considering both indoor and outdoor environments in terms of PDR, energy consumption, network lifetime, RSSI, latency, and throughput. Moreover, it also analyzes how the performance varies due to the data encryption in terms of these mentioned metrics, which would help to fill the void in this research area and help to find the most optimized configuration according to the deployment scenario.

## 3. Experimental Setup

Performance analyses were carried out in two different scenarios, indoors and outdoors. For each of the scenarios, tests were conducted with both unencrypted and AES encrypted communication for analyzing different QoS metrics—packet delivery ratio (PDR), current and energy consumption, network lifetime, link quality in terms of RSSI, latency, and throughput. Tests were performed with the commercially available XBee S2C module, which works with the Zigbee protocol at a 2.4 GHz band with 16 channels [43]. However, there are few other commercially available Zigbee modules, which are also popular among the research community for experimental analyses [44,45,46]. The specifications of these modules along with their corresponding transceiver chipset models are summarized in Table 3 below [47]:

As it is evident that even though there are a wide range of choices of Zigbee modules for different use cases, there core specifications and performances are very close to each other. XBee 3 and XBee 3 pro have higher programmable memories and can be tuned to higher Tx power, which improves the transmission range and affects the current and power consumption. However, when tuned to the same transmission power, they perform analogously with other modules, including XBee S2C [47]. For the ease of programmability, ease of integrability with sensors and microprocessors, XBee S2C is the module that this study focuses on. However, the analysis and study from this work can be generalized for the above-mentioned modules as most of those share the same specifications including the same/similar transceiver chipset. XBee S2C has an indoor range of up to 60 m, and all the tests were performed at 0–40 m distances. It can be tuned to three different transmit powers: 1 dBm, 3 dBm, and 5 dBm. Moreover, the user has the option to enable the boost mode, which would allow the modules to automatically switch to transmit power of 8 dBm when the link quality degrades. However, the boost mode is not considered in this study as it is not a stable state and longtime transmitting at this power level may damage the device. Outdoor link quality (RSSI) analysis is conducted in an open space by placing the coordinator node at one end and the end nodes at a distance ranging from 0 to 40 m from the coordinator. The outdoor experimental scenario is a parking lot allowing LoS, minimum Wi-Fi, and Bluetooth interference. For two-hop communication, the RSSI values are measured by placing one router node at an equidistance from the end node and the coordinator. For further illustration, the two-hop measurement setup for any distance is depicted in Figure 3. For multi-hop communication, Zigbee routing protocol is employed which is a modified version of ad hoc on demand distance vector (AODV). With this protocol, the node that wants to be connected initiated the communication and routes are maintained as long they are in use by the source node. Further details on Zigbee routing algorithms can be studied from [48,49,50]. For this work, the two-hop tests are conducted alongside single hop with only an RSSI measurement and round-trip latency analysis. This is because multi-hopping primarily affects these two parameters. Moreover, based on these two metrices, other metrices, such as PDR, throughput, overall energy consumption of the network, are also affected. For other tests, only single hop communication is taken into consideration. For two-hop communications, the network is configured to always choose the multi-hop for all the different distances. However, AODV routing protocol might not choose multi-hops for all the transmission distances based on network performances at that point. However, for two-hop tests with RSSI and round-trip latency analysis, multi-hopping is forced at all distances for fair comparison of single and two-hop communication. Moreover, these two tests are not focused on evaluating the routing protocol of Zigbee, rather they focus on how the performances may vary if the end node chooses two-hop instead of single hop at any given distance.

For indoor testing, lab space is taken into consideration. The RSSI values are measured by placing the coordinator at one and end nodes at a distance ranging from 0 to 40 m with NLoS. The indoor facility is divided into several rooms and a hallway with usual building blocks and metallic doors, as depicted in Figure 4. Here, E_5_, E_10,_ E_15_, E_20_, E_25_, E_30,_ E_35_, E_40_, E_45_, E_50_ are the different positions of end nodes at 5, 10, 15, 20, 25, 30, 35, 40, 45, and 50 m, respectively, from the coordinator (C_1_). We should also note that the building has extensive Wi-Fi coverage along with a large amount of Bluetooth devices, affecting the Zigbee performances in comparison to the outdoor scenario. The latency and throughput analysis are conducted by placing the coordinator and end node at 30 m from each other. These analyses evaluate how different baud rates, data encryption, number of hops, and deployment scenario (indoor/outdoor) affect QoS metrics.

## 4. Performance Analysis of QoS Metrics

This section conducts the analysis of different Zigbee QoS metrics—power consumption, packet delivery ratio (PDR), network lifetime, link quality based on RSSI, latency, and throughput and depict an overview, explanation, and evaluation of the performances based on deployment scenario and performance parameters, such as data encryption, transmission power, transmission distance, baud rates, and many hops. At first, this work analyzes the PDR and power consumption to find the optimal transmission power level. Energy consumption analysis was conducted for various transmission power levels, which led to the formulation of the equations to find the node lifetime from the current consumption captures of a data packet transmission. The link quality evaluation analyzes the variation of RSSI with distance by varying the different parameters, such as the number of hops, deployment scenarios—indoors and outdoors. It also rechecks and verifies if the RSSI performance maintains the same trend as PDR with the optimized transmission power level. Finally, by setting the transmission power to the optimal level, throughput and latency are analyzed to find the optimum baud rate, deployment scenario.

### 4.1. Evaluation of PDR for the Transmission Power Level (P_Trans_) and Energy Consumption

Multiple tests are performed with PDR and energy consumption at different P_Trans_, which allow drawing a relationship between PDR and energy consumption of a node. Thus, this analysis would also facilitate finding the optimized network lifetime and the optimized P_Trans_ where the communication is also reliable with decent PDR. To eliminate any sort of bottleneck from the hardware perspective of the modules the baud rate and bit rate are set to the maximum value supported by the modules, which are 115,200 and 250,000 b/s, respectively.

#### 4.1.1. PDR Performance at Different P_Trans_

The packet delivery ratio can be represented by Equation (2), which signifies what percentage of the sent data packet is received at the other end, a vital metric in WSN. If the PDR is bad, the network is prone to losing important data and it also increases the number of retransmissions, which eventually increases the power consumption, network traffic, and data overhead. Moreover, some IoT applications in medical sector, and industry demand higher reliability where PDR is a very crucial parameter as it also signifies the reliability of the communication [51]. For the indoor scenario, the PDR is measured by placing the coordinator at C_1_ and end nodes at E_25_, E_30_, E_35_, and E_40_, at a distance of 25, 30, 35, and 40 m from the coordinator, respectively, as presented in Figure 4. Moreover, 1000 data packets are sent from the coordinator to the end nodes at each location and from the number of received packets, PDR has been calculated. To equalize the effect of interference of other 2.4 GHz wireless technologies, three different tests at different times of the day have been carried out with the same setup and the mean of these three tests is taken into consideration. The transmission power is kept unchanged throughout the experiment and the whole process is repeated for each transmission power level, at 1, 3, and 5 dBm. For the test in the outdoor scenario, the coordinator node is placed in an open parking lot and end nodes around it with direct LoS where the interference is also minimum. Figure 5 presents the variation of PDR at different distances with three different transmission power levels.
(2)$$\mathrm{PDR}=\frac{\mathrm{Total}\;\mathrm{packet}\;\mathrm{received}}{\mathrm{Total}\;\mathrm{packet}\;\mathrm{sent}}$$

With 1 dBm, at 30 m of transmission distance, the PDR decreased to 96% and 97%, respectively, for indoor and outdoor scenarios, which are still quite reliable. The PDR with 3 dBm starts decreasing after 30 m and reaches 98% at 35 m with the indoor scenario whereas it still performs with 100% PDR in the outdoor. So, up to a transmission distance of 35 m, transmission with all the P_Trans_ perform reliably. However, the PDR performances degrade drastically at 40 m in the indoor scenario for the transmissions of all the power levels which are 94%, 86%, and 50% for 5, 3, and 1 dBm, respectively. For the outdoor scenario, the PDR performances did not degrade so drastically, 96% and 75% with P_Trans_ of 3 and 1 dBm, respectively, whereas the transmission with 5 dBm still performs with 100% PDR at a transmission distance of 40 m even. It is evident from the analysis that the decrease of the PDR at 3 dBm is not as drastic as at 1 dBm. The transmission link with 1 dBm becomes quite unreliable at 40 m for both indoor and outdoor scenarios whereas for 5 and 3 dBm it remains reasonably reliable with decent PDR indoor and performs with even better PDR outdoor. To find an optimized P_Trans_ level with decent PDR and energy consumption, and extensive current and energy consumption analysis is done. This would help to realize the trade-offs between P_Trans_, PDR, and energy consumption and to find an optimized P_Trans_ where it can perform both reliably and energy efficiently.

#### 4.1.2. Energy Consumption and Network Life at Different P_Trans_ Levels

To analyze the energy consumption better, it is important to understand how the communication is performed, thus a data packet is transferred from a coordinator to an end node or vice versa. The Zigbee protocol is based on IEEE 802.15.4 and Zigbee follows this standard completely for medium access control (MAC) and the physical (PHY) layer. However, it is modified and different than that of IEEE 802.15.4 in the network (NWK) and application (APS) layer where it allows the Zigbee to form a mesh network and enable multi-hop communication from the end node to the coordinator. However, the energy consumption of a sensor network heavily depends on the PHY and MAC layer, which are the same as the IEEE 802.15.4. Zigbee follows the CSMA/CA to send data from one node to another.

In this experiment, data packets of 30 bytes are sent from the coordinator to an end node. For measuring the current consumption and the energy of the total packet reception, the end node is powered up from a 3.5 DC supply and connected to an oscilloscope across a shunt resistor of 9 Ω. This setup is depicted by Figure 6a, and Figure 6b presents the current consumption during the successful reception of a packet. To send a data packet, at first, the coordinator broadcasts a beacon message in its network [52]. The end nodes are configured in cyclic sleep mode. They wake-up after the predefined sleep time and listen for the beacon message from the sender, which is the coordinator in this case. To wake-up from sleep and receive the broadcasted beacon from the coordinator, the receiver takes some time. During this wake-up time, the receiver stays in radio idle mode as presented by annotation 1, whereas annotation 2 denotes the reception of this beacon message. After that, the receiver sends a data request to the coordinator. After receiving a broadcast beacon, the receiver stays in radio standby mode as annotation 3 until it sends the data request to the coordinator as presented by annotation 4. Then the receiver radio goes to idle mode and stays in idle mode until it receives the acknowledgment of the data request. After receiving the data request from the receiver, the sender (coordinator) waits for a backoff time according to the predefined contention window and then performs the clear channel assessment (CCA) before sending the data. The receiver goes to the receiving mode after the reception of this acknowledgment from the coordinator and remains in that mode until the data transfer is over as denoted by annotation 5. It processes the received data and upon successful reception of the data it sends an acknowledgment back to the coordinator, which is denoted by annotation 6 and then goes to sleep mode again. The receiver can also wake-up upon receiving the broadcast beacon from the coordinator or depending on the configuration, it can follow any predefined sleep time to wake-up at a regular interval. All the different stages of the packet reception by an end node and annotation of Figure 6b are summarized in Table 4.

The duration of these data reception stages varies according to the P_Trans_ level and the packet size. Moreover, the variation of the duration of these stages affects the overall energy consumption of the nodes, thus energy efficiency and network lifetime. The current consumption is also increased by the data encryption, which contributes to decreasing the energy efficiency. The current consumption captures of the packet reception at different P_trans_ levels with both 128-bit AES encryption and without encryption are presented in Figure 7. We should also note that the end node can receive the broadcast beacon from the coordinator at any time during its wakeup duration. Thus, the duration of the first idle time (before receiving the beacon broadcast) as depicted by annotation 1 (first one before annotation 2) of Figure 6b may vary randomly. To normalize this effect for each of the instances, a mean of 20 readings is taken into consideration. Figure 7 shows that the current consumption during transmitting, and reception of a packet is highest for P_Trans_ level of 5 dBm, which is followed by 3 dBm, and the current consumption with a transmission power of 1 dBm is the lowest. Moreover, it is evident from Figure 7 that the difference of the current consumption with different transmission power levels mostly differs in transmit and reception mode. The current consumption during radio standby time, idle time, and sleep time is almost the same for all the P_Trans_ levels. Moreover, with AES encryption the current consumption during transmits and reception is slightly higher than that of unencrypted data due to the higher time and resource requirement of the encryption procedure. The exact current and energy consumption during each stage of the data reception for all the P_trans_ levels and both encrypted and unencrypted communication are depicted in Table 5 and Table 6, respectively. The mean of the 20 readings is taken into consideration for the data collection.

The energy consumption for each stage of the transmission is calculated with Equation (3) where *I* is the average current consumption, *Rsh* is the shunt resistor of 9 Ω across which the XBee module is connected, and *T* is the duration of that stage.
(3)$$\mathrm{Energy}\;\mathrm{Consumption}=I^{2}R_{sh}T$$

From Table 5 and Table 6, we can see that for both the encrypted and unencrypted communication, the energy consumption is highest with a transmission power level of 5 dBm and the lowest with 1 dBm. As encrypted communication performs functions, such as encrypting the data before transmission and decrypting the data after reception, it consumes more energy for all the transmission levels in comparison with unencrypted communication. However, provided the security that the AES encryption provides to the communication, this increased energy consumption for the encryption is considerable and is recommended for most indoor and outdoor IoT applications. The energy consumption is highest in the data reception stage, which is followed by the radio standby stage for all the mentioned cases. One of the interesting facts of both encrypted and unencrypted communication is that the difference of the total energy consumption between a 1 and 3 dBm P_Trans_ level is very little in comparison with the energy consumption difference between 3 and 5 dBm. In fact, with encrypted communication, this difference of energy consumption between P_Trans_ levels of 1 and 3 dBm is almost inconsiderable. On the contrary, if PDR for different P_Trans_ levels is analyzed, it is noticeable that PDR improved significantly from 1 to 3 dBm. As with 1 dBm, at 40 m, the PDR drops drastically to 75% and 50%, respectively, for outdoor and indoor environments; it is beyond consideration.

With 3 dBm, the PDR is 100% up to a transmission distance of 35 and 30 m, respectively, for outdoor and indoor environments. Moreover, this drops to 96% and 86% for outdoor and indoor environments, respectively, at a transmission distance of 40 m, which is decent for most IoT applications. As with very little increase of energy consumption from 1 to 3 dBm, the PDR improves radically, the 3 dBm transmission power level is recommended by authors for network optimization. To improve the PDR further with a 5 dBm P_Trans_ level, the energy consumption increase will significantly decrease the overall network life dramatically. Thus, 3 dBm is chosen for the rest of the study and analysis. The impact of the transmission power level on the network lifetime would be much more obvious by analyzing the network lifetime of the considered XBee S2C module. The network lifetime of any node can be calculated from the current consumption and duration of each stage of data transmission, which is presented in Table 5 and Table 6. For calculating the network lifetime of any node, we formulated Equations (4)–(7) where *T_active_* is the total duration, *I_active_* is the total consumed current during the transmission of a single packet. The XBee module is configured to go to sleep immediately after the data transmission and wake-up after the predefined sleep time for the next transmission. The duration between two consecutive data transmissions is sleep time, which is configured by the user. The total number of transmissions in any given time depends on this user-defined sleep time, which is denoted by *T_sleep_*. The current consumption during *T_sleep_* is minimum, which is 0.6 mA for the XBee S2C module and is presented by *I_sleep_* in Equation (6). The total lifetime of a node is calculated by Equation (7) and is presented with *T_lifetime_* where *B_capacity_* is the capacity of the DC source of the node in mAh [53]. The node lifetime is calculated for a battery of 5000 mAh capacity with different *T_sleep_* intervals for a comparative analysis of different transmission power levels of 1, 3, and 5 dBm. The network lifetime for different sleep intervals, i.e., packet intervals and transmission power levels with a packet size of 30 bytes and battery capacity of 5000 mAh, is presented in Figure 8.
(4)$$T_{active}=T_{tx1}+T_{tx2}+T_{rx1}+T_{rx2}+T_{idle}+T_{sb}$$
(5)$$I_{active}=\frac{T_{tx1}T_{tx1}+T_{tx2T_{tx2}}+T_{rx1T_{rx1}}+T_{rx2}T_{rx2}+T_{idle}T_{idle}+T_{sbT_{sb}}}{T_{active}}$$
(6)$$I_{drain}=\left(\frac{T_{active}}{T_{sleep}}\times I_{active}\right)+\left(1-\frac{T_{active}}{T_{sleep}}\right)\times I_{sleep}$$
(7)$$T_{lifetime}=\frac{\frac{B_{capacity}}{I_{drain}}}{365\times 24}$$

The lifetime of an XBee S2C module can range between 0.93 and 0.94 years with a packet size of 30 bytes, battery size of 5000 mAh, and packet interval of 60 s. Thus, it performs considerably energy efficiently with Zigbee protocol in comparison with the other contemporary technologies, such as Z-wave, Bluetooth low energy (BLE), and LoRa. The node lifetime with unencrypted communication is slightly higher than that of encrypted communication but it is very insignificant considering the security and data privacy AES encrypted communication provides. Focusing on the node lifetime estimation of encrypted communication, the lifetime with transmission levels of 1 and 3 dBm are almost the same whereas the node lifetime decreases with a higher transmission power of 5 dBm. Even though this difference is not very significant, the node lifetime degrades significantly with 5 dBm in comparison with 3 dBm with the decrease of the packet interval and packet size. Thus, for task-intensive applications, a transmission power level of 3 dBm is preferred. Moreover, with 3 dBm, the performance, such as PDR, is quite decent as mentioned earlier, which does not improve drastically with 5 dBm. As it is analyzed, 3 dBm is the most optimized transmission power level considering the trade-off of PDR and network lifetime, the rest of the evaluation with RSSI, latency, and data throughput focuses mainly on 3 dBm, 128-bit AES encrypted communication.

### 4.2. RSSI Analysis for Indoor and Outdoor Multi-Hop Communication

RSSI is one of the metrics used to determine the link quality between the transmitter and the receiver. Understanding this metric is essential to having a clear idea about the transmission range and sensitivity of the deployed network. This would facilitate to better localize the nodes in the network with energy optimization and reliability. The comparative study of the parameters, such as a signal to interference and noise ratio (SINR), link quality indicator (LQI), along with RSSI, would have given better understanding of the overall link quality. However, this study focuses only on the RSSI readings to analyze the link quality. Along with the other factors, RSSI depends primarily on the transmission power level and obstruction between the sender and receiver, transmission distance, and interference. Thus, it is expected to have varied RSSI readings with the change of transmission power, the number of hops, the distance between the end node and coordinator, and the deployment environment.

For RSSI measurement in the indoor scenario, the nodes are deployed in an office environment as described in Figure 4. The coordinator is placed at position C_1_ and RSSI readings are taken by deploying the end nodes at 0, 5, 10, 15, 20, 25, 30, 35, and 40 m from the coordinator corresponding to the C_1_, E_5_, E_10_, E_15_, E_20_, E_25_, E_30_, E_35_, and E_40_ annotation of Figure 4. We should note that for all of the cases no line of sight was maintained with obstructions, such as a concrete wall, lab equipment, and machinery. For an outdoor environment, the coordinator is placed at the center of an open parking area and the end nodes around it with distances ranging from 0 to 40 m with a direct line of sight for all the cases. At every point of measurement for all the instances, a mean of 50 readings is taken into consideration. Figure 9 presents the link quality of transmission in terms of RSSI with different transmission power levels, distances, and deployment environments. In both the deployment scenarios: indoors and outdoors, the link quality with the transmission power level of 5 dBm is the best and that of 1 dBm is the worst. In the indoor office environment, the values fluctuate more in comparison to the outdoor scenario, due to the NLoS transmission medium. However, it follows the same trend for all the transmission power levels, which ranges from −40 to −92 dBm for P_trans_ = 5 dBm and from −40 to −97 dBm for P_trans_ = 1 dBm in the indoor scenario. Comparatively, the link quality is better in the outdoor scenario with RSSI ranging from −35 to −71 dBm for P_trans_ = 5 dBm and from −36 to −82 dBm for P_trans_ = 1 dBm. The link of the outdoor scenario is better than that of the indoor office environment as it offers an LoS transmission medium with less interference and no obstruction in between. We should note that the RSSI performances of both the indoor and outdoor environments are good enough in comparison to the receiver sensitivity of the XBee S2C module of −100 dBm. It would be more interesting to analyze the RSSI measurements of multi-hop communication in comparison with the single hop. For two-hop communication, one router node is placed at an equidistance from the coordinator and end node as presented in Figure 3. To analyze and compare the link quality of single-hop and two-hop communication for both indoor and outdoor scenarios, the study focuses only on the transmission power level of 3 dBm as it is most optimized. Figure 10 depicts the RSSI values of single-hop and two-hop communication with P_Trans_ = 3 dBm for various transmission distance and deployment scenarios. In an indoor environment, the RSSI improves by 14.73% at 40 m, which is 10.90% at 5 m of transmission distance while we switch to two-hop communication from a single hop. Similarly, in an outdoor scenario, two-hop communication improves the RSSI by 7.89% at a transmission distance of 40 m, which is 6.38% at 5 m. As the multi-hop communication improves the link quality significantly for both indoor and outdoor environments it facilitates increased transmission distance from the coordinator to the end node, as well as improves reliability. Thus, in such a scenario where LoS is minimum, and interference is high, multi-hop communication is preferred. The multi-hop mesh network also covers a much larger area in comparison with single-hop communication, thus it contributes to improving the scalability of the network.

### 4.3. Latency Analysis of Zigbee with Multi-Hop AES Encrypted Communication

Latency is another important aspect of IoT and sensor networks. It is more crucial in real-time applications, such as health monitoring and vehicular communication [51]. Latency is affected by various factors including baud rates, data packet size, number of hops, and encryption of the data. Based on the real-time data collection, this work conducts the experimental analysis on how the above-mentioned three factors affect the latency of Zigbee communication. Baud rates being one of the primary and obvious factors affecting the latency, this work first studies this. For measuring the latency at different baud rates, the round-trip latency (RTL) is taken into consideration by sending packets ranging from 10 to 60 bytes from the coordinator to the end node, which is again looped back to the coordinator as presented in Figure 11. RTL is the summation of transmit time (T_TX_) and reception time (T_RX_). The test is conducted in an indoor environment as preliminary tests suggest no significant variation in latency measurement in indoor and outdoor environments. The coordinator is placed at C_1,_ and the end node is deployed with a 30 m distance at E_30_ as depicted by Figure 4. The Rx and the Tx pin of the end node module are shorted to loop back the received data to the coordinator without any data preprocessing delay. Figure 12 depicts the latency of single-hop Zigbee communication with a variation of baud rates and packet size for both encrypted and unencrypted communication with a transmission distance of 30 m. The mean of 50 readings is taken into consideration for any data point of the plots. For both AES encrypted and unencrypted data, the latency increases linearly with the increase of data packet size for all the baud rates. Overall, with any packet size, RTL is minimum with the baud rate of 115,200, which increases with the decrease of the baud rate and reaches the maximum at 4800. With the baud rate of 115,200, the RTL is 0.075 s and 0.028 s for the packet size of 10 bytes, which increases linearly and reaches 0.095 s and 0.04 s for the packet of 60 bytes for encrypted and unencrypted data, respectively. Thus, the latency increases by 5.34% and 8.57%, respectively, for encrypted and unencrypted data with the increase of each 10 bytes of the packet size. On the other hand, when the transmission is switched from the baud rate of 115,200 to 4800, with every 10 bytes/packet, the RTL increases by 66.67% and 221.43%, respectively, for encrypted and unencrypted communication, which is 257.89% and 625%, respectively, for 60 bytes/packet. Thus, it is evident that the trend of the percent increase of the RTL with the increase of packet size and also with the decrease in the baud rate is stepper in unencrypted communication in comparison to the AES encrypted one. Now, it is quite intriguing to analyze how the latency changes due to encryption and the hopping of the data. To do this study, the best performing baud rate, 115,200, is taken into consideration for one-hop and two-hop transmission. This would greatly help to understand better the trade-offs of AES and multi-hop communication. For this, the coordinator is placed at C_1_, the end node at E_30_, which is at a 30 m distance from the coordinator as depicted by Figure 4. For two-hop communication, a router is placed at an equidistance from a coordinator and the end node. Figure 13 compares the RTL performances of the one-hop and two-hop routing with and without AES encryption. It is perceptible from the figure that the measured RTL of one-hop communication both with and without AES encryption is significantly less than that of two-hop communication instances. The RTL of one-hop communication with and without encryption at 10 bytes/packet is 50% and 186.27% less than that of the corresponding two-hop communications and with 60 bytes/packet, it is 319.58% and 517.43% less than that of two-hop with and without AES communication, respectively.

Therefore, even though the increased number of hops increases the reliability in terms of link quality and delivery ratio and also increases the transmission range, it also significantly increases the latency of the communication. Therefore, deploying more router nodes increases the reliability, PDR, and transmission range, but affects the latency performance drastically. Thus, multi-hop communication is not suitable for real-time applications where a delay is very critical. We can also analyze that the encryption also worsens the latency performances if not as worse as the increase of the number of hops. The 128-bit AES encryption in Zigbee increases the latency by 153.52% and 134.87% for 10 bytes/packet and 60 bytes/packet, respectively, for one hop and the increment is 32.84% and 59.35%, respectively, for two-hop.

Thus, as the number of hops increases the effect of encryption on latency lessens. This is an interesting fact that in a larger network with a higher number of hops, the encryption by itself will less affect the latency in comparison to the single-hop communication. These are the trade-offs of the higher number of hops, data encryption, and the latency that need to be considered while designing a Zigbee network for any indoor or outdoor applications.

### 4.4. Analysis of Throughput Considering Data Encryption and Deployment Scenario

The baud rates define how fast the data transfers, i.e., the data transfer rate through serial ports. Thus, it is one of the primary and important factors of the overall throughput of communication. However, other parameters, such as the size of the packets and the deployment scenario, which controls the transmission medium and LoS, may limit the data throughput significantly. The data throughput of the Zigbee protocol can reach up to 250 Kbps depending on the number and types of nodes, transmission medium, operating frequency, and allocated bandwidth [54]. However, it greatly varies and is reduced down to 4.5–5 Kbps in a real-world scenario for point-to-point communication with actual transceivers as presented by [55]. This study by Piyare et al. considered only the baud rates and packet size as the performance parameters of throughput. However, it is also important to understand how the throughput changes with deployment scenario and the data encryption. Thus, our work performed these studies and presents a comprehensive analysis of throughput. For indoor scenarios, the study was performed by deploying the coordinator node at C_1_ and the end node at E_30_, which was at 30 m distance from each other as depicted by Figure 4. For the outdoor study, the deployment scenario was the same, but it always maintained a direct LoS and was implemented in an open parking lot to experience usual outdoor interference. Figure 14 presents the throughput vs. packet size at three different baud rates—9600, 57,600, and 115,200 for two different deployment scenarios. The study shows that the throughput performances of non-encrypted communication are slightly better than that of encrypted communication for both indoor and outdoor deployment. This is because of the increased data overheads due to encryption and also because of the additional time, it takes for encrypting and decrypting the data, which is evident from the current consumption and latency study as presented earlier in Section 4.1.2 and Section 4.3. On the other hand, transmission throughput is highest for the highest possible baud rate of 115,200 with the module XBee S2C for all the different packet sizes and deployment scenarios and it is lowest for the lowest considered baud rate of this study, which is 9600. Thus, the highest possible baud rate that is supported by the network devices is always preferred. For 10 and 60 bytes/packet with 115,200 bps, AES encryption decreases the throughput by 13.92% and 3.84%, respectively, for indoor deployment, which is 8.89% and 2.77%, respectively, for outdoor deployment. It is evident from the analysis that the percent decrease in throughput due to encryption is significantly improved for larger packet sizes for both scenarios. This happens as the data overhead for the encryption remain the same even though the encrypting and decrypting times vary with the variations of the packet sizes. It is another important aspect and interesting topic to study how the throughput is affected by the indoor deployment i.e., the non-LOS medium, multi-path propagation, and interference of the co-existing technologies, such as Wi-Fi and other RF technologies on the same ISM band. The throughput increases by 8.89% and 3% for 10 and 60 bytes/packet, respectively, when the network deployment scenario switches from indoors to outdoors for non-encrypted communication, which is 13.92% and 4.08% for encrypted communication. However, the overall throughput of the network can be increased significantly by deploying more router nodes, but this will increase the latency and the overall network cost. This is the tradeoff of the throughput and latency, which plays a great role in Zigbee network designing.

However, these QoS metrics, especially the network lifetime, and energy efficiency, can be improved significantly by adopting the data aggregation approach at the end nodes based on correlation among the sensor data as demonstrated by [56,57]. This can be a great solution to improve the performance metrics as the data recorded by the closer nodes are correlated; thus, the total payload can be cut down significantly by compressing the data upon aggregation. Moreover, many researchers have also suggested the time division multiple access (TDMA)-based approach with Zigbee instead of traditional CSMA/CA, which has shown great potential in achieving better performances in terms of throughput and energy consumption [58,59,60]. However, this approach may encounter problems due to inefficient clock synchronization, which can be pro-actively addressed by a priority-based direction aware media-access control protocol by Abbas et al. [61]. The authors aimed to propose an energy-efficient, optimized, and improved Zigbee protocol based on the performance analysis of this study.

## 5. Conclusions

In this work, a comprehensive performance analysis of single and multi-hop Zigbee communication was performed based on the real-world indoor and outdoor testbeds and measurements where the XBee S2C module was used as the transceiver. Performance metrics, such as PDR, power consumption, link quality, latency, and throughput were analyzed with the variations in transmission power level, transmission distance, packet size, baud rates, deployment scenario, and data encryption. This work also drew out the trade-offs and performance limitations of these metrics, which would benefit both academia and the industry for designing and deploying Zigbee in both indoor and outdoor applications. The study shows that the transmission power level of 3 dBm is the most optimized one in terms of PDR and energy consumption, i.e., the node lifetime. Node life can reach up to 0.94 years with a battery of 5000 mAh, transmitting 30 bytes/packet in each minute. Moreover, 3 dBm transmissions show promising performances, also in link quality, maintaining −95 and −82 dBm for single and two-hop, respectively, in a compacted office environment with a transmission distance of 40 m. Even though multi-hop communication can increase the transmission radius significantly, it can also increase the latency up to 517.43%. On top of it, AES 128-bit encryption can degrade the latency up to 153.52%. These are the trade-offs of multi-hop and encrypted communication that one must consider while designing the network. The analysis also presented the improvement of the throughput performance with higher baud rates, better LoS, less interference, and higher packet sizes.

We should note that the experiment that was conducted in this study was based on the Digi XBee S2C module, and the performance may vary slightly with the change of transceiver modules.

## Figures and Tables

**Figure 1 sensors-22-03245-f001:**
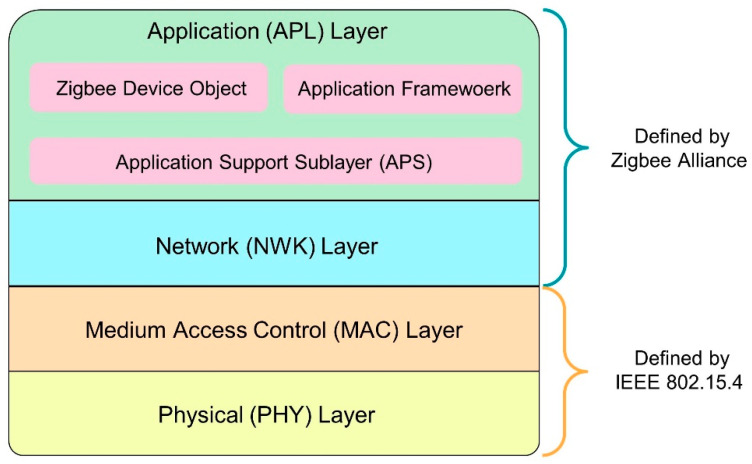
Zigbee protocol stack.

**Figure 2 sensors-22-03245-f002:**
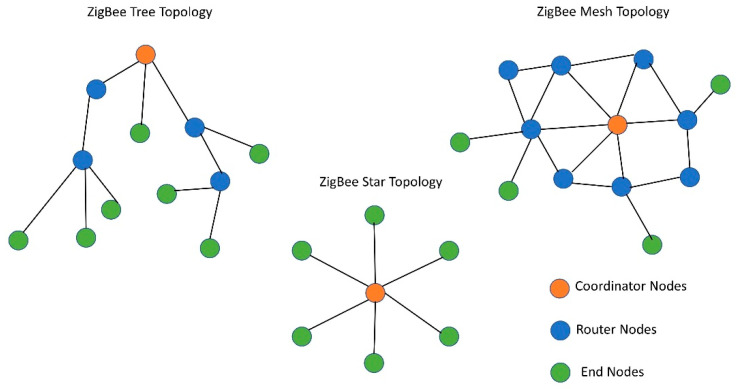
Different topologies of the Zigbee standard.

**Figure 3 sensors-22-03245-f003:**
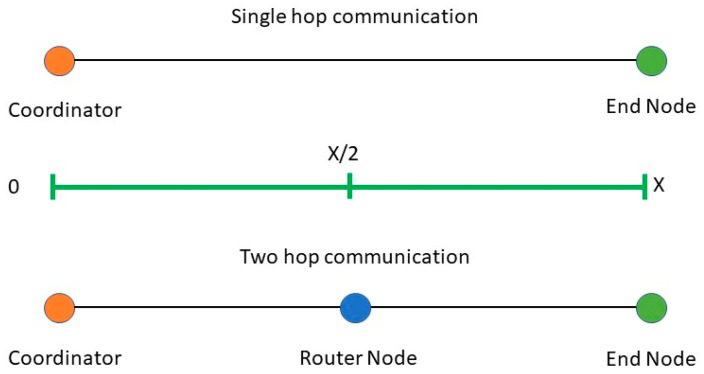
Single hop and two-hop setup for QoS measurements.

**Figure 4 sensors-22-03245-f004:**
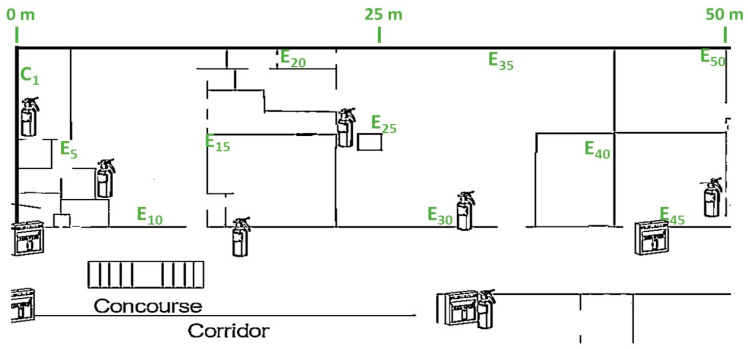
Location of the coordinator end nodes in the indoor lab environment.

**Figure 5 sensors-22-03245-f005:**
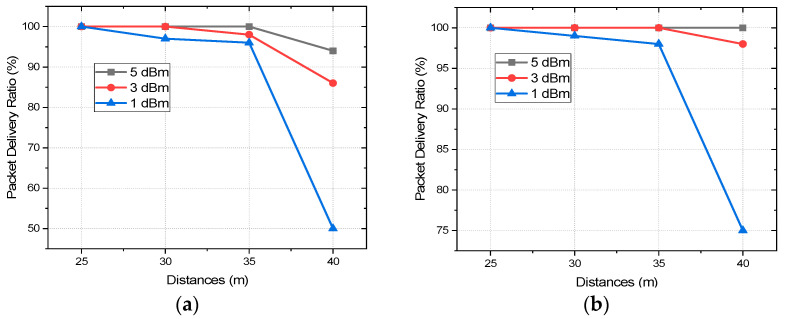
PDR vs. distances for different transmission power levels in (**a**) indoor and (**b**) outdoor environments.

**Figure 6 sensors-22-03245-f006:**
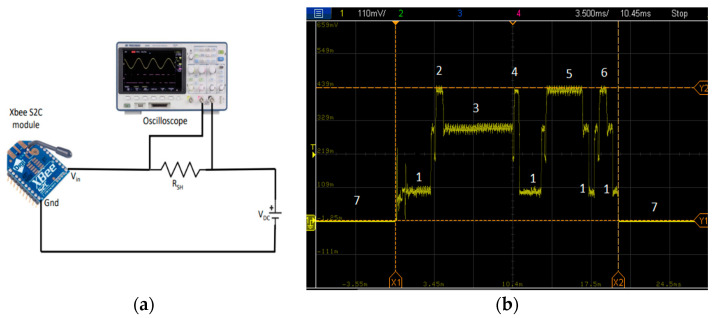
(**a**) Measurement setup and (**b**) current consumption capture of the successful reception of a packet.

**Figure 7 sensors-22-03245-f007:**
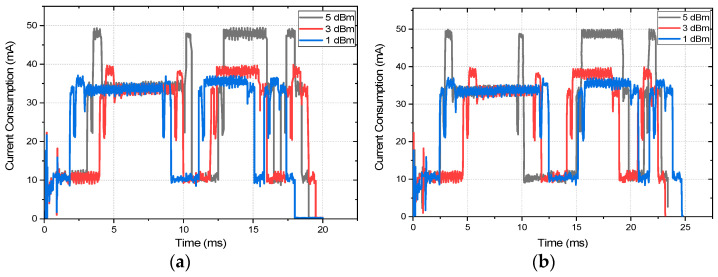
The current consumption captures at different P_trans_ levels (**a**) without encryption and (**b**) with 128-AES encryption.

**Figure 8 sensors-22-03245-f008:**
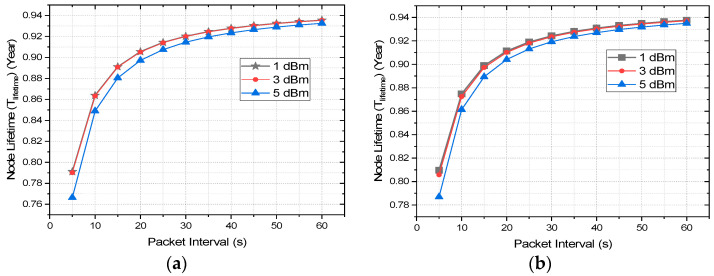
End node lifetime estimation of XBee S2C with 5000 mAh battery at various packet intervals: (**a**) AES encrypted communication; (**b**) unencrypted communication.

**Figure 9 sensors-22-03245-f009:**
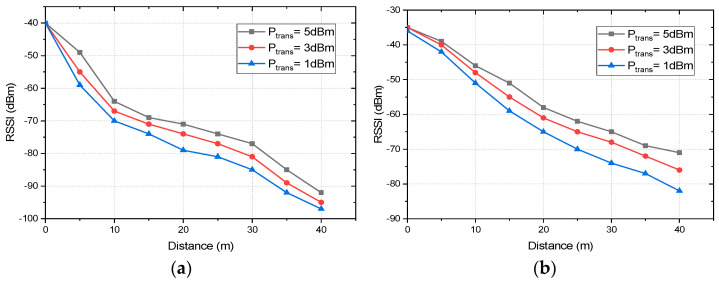
The link quality of transmission in terms of RSSI with different transmission power levels, distance, and deployment environment: (**a**) indoors and (**b**) outdoors.

**Figure 10 sensors-22-03245-f010:**
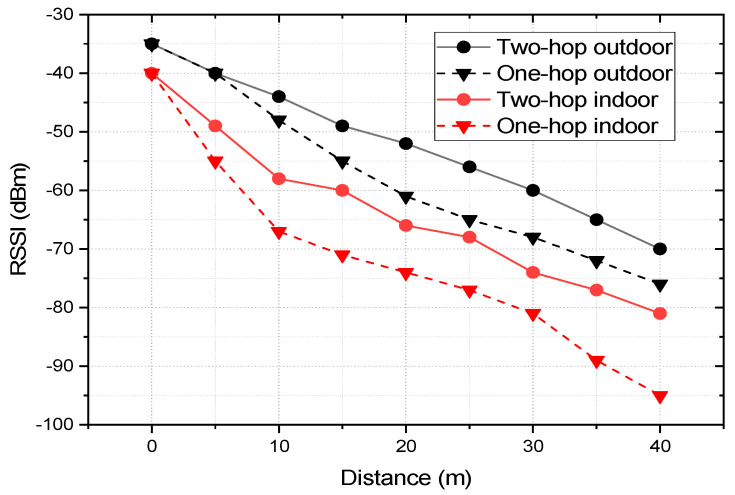
RSSI values of single-hop and two-hop communication with P_Trans_ = 3 dBm for various transmission distances and deployment scenarios.

**Figure 11 sensors-22-03245-f011:**
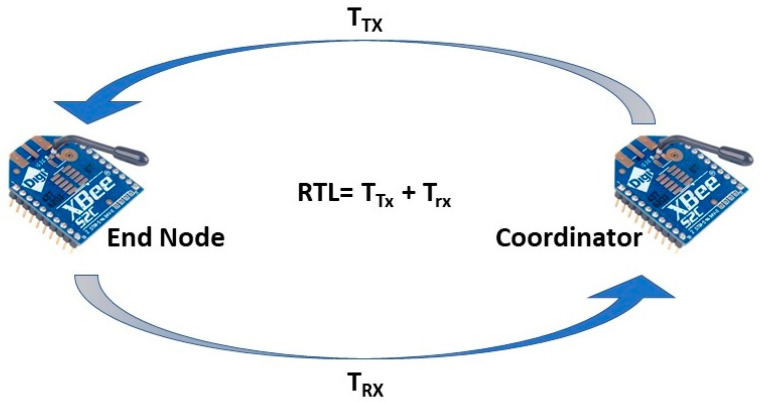
RTL measurement setup for latency analysis.

**Figure 12 sensors-22-03245-f012:**
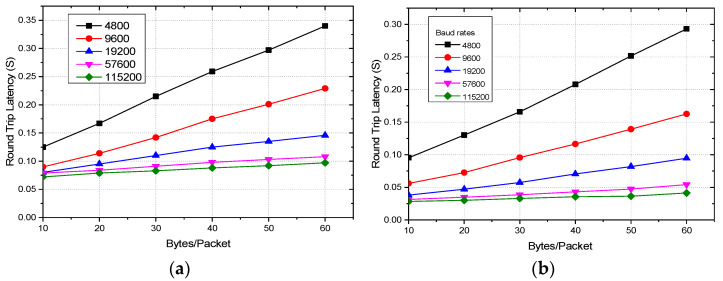
Zigbee round trip latency with variations of baud rates and packets size for (**a**) encrypted and (**b**) unencrypted communication.

**Figure 13 sensors-22-03245-f013:**
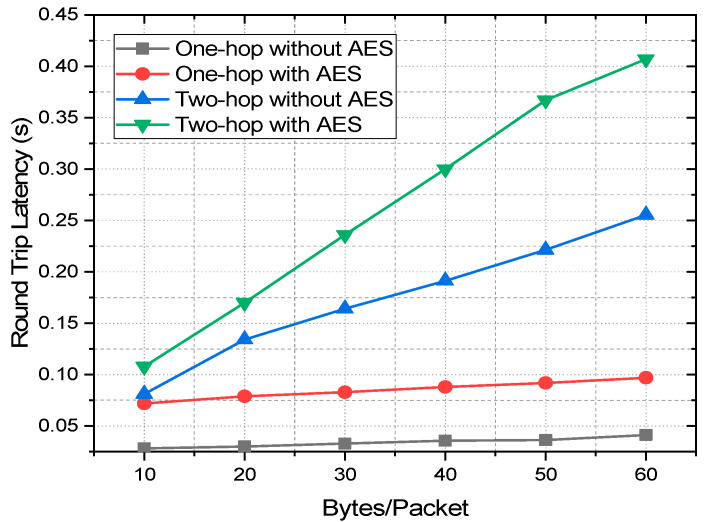
RTL performances of one-hop and two-hop communication with and without AES encryption.

**Figure 14 sensors-22-03245-f014:**
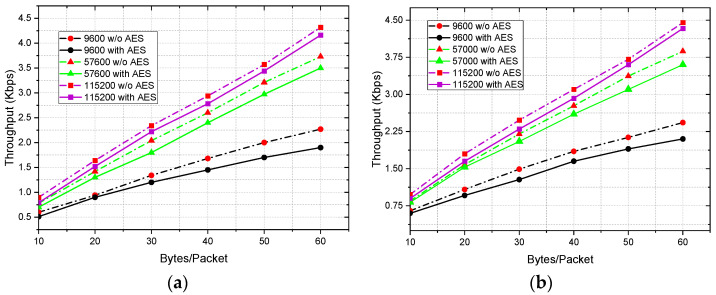
Throughput vs. packet size at three different baud rates for (**a**) indoor and (**b**) outdoor deployment.

**Table 1 sensors-22-03245-t001:** PHY, MAC, and NWK layer characterization of Zigbee [9].

Networking Layer	Parameters	Zigbee Characterization
PHY Layer	Frequency Band	2.4 GHz, 915 MHz, 868 MHz,
	Throughput	250 Kbps for 2.4 GHz
		40 Kbps for 915 MHz
		20 Kbps for 868 MHz
	Modulation	BPSK, O-QPSK
	Tx Power [16]	−3 to 10 dBm
	Minimum receiver Sensitivity	−85 dBm
	Physical Channels	16 channels: 2.4 GHz
		10 Channels: 915 MHz
		1 Channel: 868 MHz
	Channel Bandwidth	2 MHz
MAC Layer	Multiple Access Scheme	CSMA-CA, Slotted CSMA-CA
	CRC length	2 bytes
	Identifiers	16-bit short address
		64-bit long address
NWK Layer	Network Topology	Star, Tree, Mesh, Point-to-Point
	Hopping	Single and Multi-hop
	Device Type/Mode	Coordinator, Router, End Device
	Networking Technology	PAN

**Table 2 sensors-22-03245-t002:** Summary of the research focus, QoS metrics, and limitations of the discussed works.

Research	Deployment Scenario	QoS Parameters	Limitation
Evaluation of Zigbee topology [17]	Simulation	Throughput, end-to-end delay	Only focused on throughput and delay; real-world networking performance might vary from simulation; encrypted communication is not considered.
Data transmission performance analysis with XBee Pro 2B [32]	Indoor and outdoor	Transmission range	Other QoS parameters, such as throughput, link quality, latency, and power consumption were not considered; encrypted communication was not considered.
Zigbee performance analysis in Various Environments [19]	Indoor LOS and NLOS	Delay, throughput, packet loss	Comparative performance of encrypted communication and variations of deployment scenarios were not addressed.
Comparative study of Zigbee topologies [41]	Simulation	Latency, throughput, packet loss, and energy consumption	Did not consider encrypted communication.
Performance Evaluation of Zigbee [33]	Simulation	Delay, power consumption	Parameters, such as throughput, link quality, and data encryption were not considered.
Analysis of Zigbee data transmission [34]	-	Latency, packet loss, throughput	RSSI, energy consumption, data encryption, and NLoS scenario were not considered.
Performance evaluation of Digi Mesh and Zigbee mesh [35]	-	Throughput, round trip time, RSSI, routing recovery time	Energy consumption, data encryption, and different deployment scenarios were not considered.
Performance of Zigbee network topologies [42]	Simulation	Throughput, PDR, latency, energy consumption, security	Based on simulation, which might differ from the deployed network performance.
Performance analysis of Zigbee large scale network [36]	Simulation	Latency, throughput	Did not consider data encryption and other QoS parameters.
Performance analysis of Zigbee WSN [37]	Simulation	Throughput, delay, data traffic	Did not consider other performance metrics, such as RSSI and power consumption.

**Table 3 sensors-22-03245-t003:** Technical specifications of the popular commercial Zigbee modules.

Zigbee Module	Transceiver	Programmable Memory	Programmable CPU Clock	No. of Channels	Receiver Sensitivity	Tx Power	Tx and Rx Current
XBee S2C	Silicon Labs EM357 SoC	32 KB Flash/2 KB RAM	Up to 50.33 MHz	16	−100 dBm/−102 dBm (boost mode)	3.1 mW (+5 dBm)/6.3 mW (+8 dBm) boost mode	Tx: 33 mA @ 3.3 VDC/45 mA boost mode Rx: 28 mA @ 3.3 VDC/31 mA boost mode
XBee-Pro S2C	Silicon Labs EM357 SoC	32 KB Flash/2 KB RAM	Up to 50.33 MHz	15	−101 dBm	63 mW (+18 dBm)	Tx: 120 mA @ 3.3 VDC Rx: 31 mA @ 3.3 VDC
XBee S2D	Silicon Labs EM3587 Soc	N/A	N/A	15	−100 dBm/−102 dBm (boost mode)	3.1 mW (+5 dBm)/6.3 mW (+8 dBm) boost mode	Tx: 33 mA @ 3.3 VDC/45 mA boost mode Rx: 28 mA @ 3.3 VDC/31 mA boost mode
XBee 3	Silicon Labs EFR32MG SoC	1 MB/128 KB RAM	-	16	−103 dBm normal mode	+8 dBm	Tx: 40 mA @ 8 dBm Rx: 17 mA
XBee 3 Pro	Silicon Labs EFR32MG SoC	1 MB/128 KB RAM	-	16	−103 dBm normal mode	+19 dBm	Tx: 135 mA @ 19 dBm Rx: 17 mA

**Table 4 sensors-22-03245-t004:** Data reception stages of an end device.

Annotation of Figure 6b	Stages of Data Transmission/Reception	Brief Explanation
1	Idle time (T_idle_)	Node is active, but the radio is not active. The nodes tend to stay in idle mode to save energy.
2	Data reception time (T_rx_)	Reception of the beacon message broadcasted from a coordinator.
3	Radio standby time (T_sb_)	Radio stays in standby mode before sending a data request to a coordinator or any sender as it waits for backoff time and performs CCA.
4	Data transmit time (T_tx_)	End node sends the data request to the coordinator/sender
5	Data reception time (T_rx_)	End node receives the ACK of the data request and goes to receiving mode and waits until data transmission is over
6	Data transmit time (T_rx_)	ACK is sent upon successful reception of the packet.
7	Sleep Time (T_sleep_)	End node remains in the sleep mode before and after the reception of the data packet as defined by the experiment configuration

**Table 5 sensors-22-03245-t005:** Current and energy consumption during each stage with encrypted communication.

Annotation of Figure 6b	Stages of Data Transmission/Reception	P_Trans_								
		1 dBm			3 dBm			5 dBm		
		Duration (ms)	Avg Current Consumption (mA)	Energy Consumption (μJ)	Duration (ms)	Avg Current Consumption (mA)	Energy Consumption (μJ)	Duration (ms)	Average Current Consumption (mA)	Energy Consumption (μJ)
1	Idle time (T_idle_)	7.3	10.5	7.29	10.0	10.75	10.4	10.75	10.75	11.18
2	Data reception time (T_rx1_)	1.0	36.0	11.66	1.05	38.8	14.22	1.0	49.0	21.60
3	Radio standby time (T_sb_)	5.4	34.0	56.18	5.2	34.0	54.1	6.25	34.0	65.02
4	Data transmit time (T_tx1_)	0.8	35.3	8.97	0.8	38.4	10.61	0.8	49.8	17.85
5	Data reception time (T_rx2_)	6.0	35.0	66.15	5.0	37.5	63.28	4.8	47.5	97.47
6	Data transmit time (T_tx2_)	2.5	34.5	26.78	2.3	36.0	26.82	2.3	41.0	34.79
Total energy consumption (μJ)				177.03			179.43			247.91

**Table 6 sensors-22-03245-t006:** Current and energy consumption during each stage with unencrypted communication.

Annotation of Figure 6b	Stages of data Transmission/Reception	P_Trans_								
		1 dBm			3 dBm			5 dBm		
		Duration (ms)	Average Current Consumption (mA)	Energy Consumption (μJ)	Duration (ms)	Average Current Consumption (mA)	Energy Consumption (μJ)	Duration (ms)	Average Current Consumption (mA)	Energy Consumption (μJ)
1	Idle time (T_idle_)	6.1	10.5	6.05	8.4	10.5	8.33	6.8	10.5	6.74
2	Data reception time (T_rx1_)	0.9	36.0	10.49	0.9	38.2	11.81	0.9	47.9	18.58
3	Radio standby time (T_sb_)	6.0	33.8	61.69	5.5	33.7	56.21	5.5	33.9	56.88
4	Data transmit time (T_tx1_)	0.7	35.1	7.76	0.7	36.0	8.16	0.7	47.5	14.21
5	Data reception time (T_rx2_)	4.0	35.0	44.10	4.0	36.0	46.65	4.8	46.0	91.41
6	Data transmit time (T_tx2_)	2.1	33.9	22.23	2.1	36.0	25.07	2.1	40.0	30.24
Total energy consumption (μJ)				152.32			156.23			218.06

## Data Availability

Not applicable.

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
