# Peer review of "Comprehensive Performance Analysis of Zigbee Communication: An Experimental Approach with XBee S2C Module"

_sensors, 2022, doi:10.3390/s22093245_

Round 1
Reviewer 1 Report
This paper presents a detailed measurement on the ZigBee power consumption under different situations. Based on this, the authors further introduce the solutions to evaluate the most optimized transmission power level, network lifetime and the trade-offs.
Overall, the paper is interesting with solid evaluation. The authors could consider the following to further improve the paper.
[1] Discussion about other ZigBee modules. This paper focuses on Xbee, lacking a brief analysis/discussion about other ZigBee nodes. A natural question is the generality of the proposed mechanism.
[2] A larger ZigBee network with more hops could be tested, if possible.
Author Response
Please see the updates listed in the attached document.

Reviewer 2 Report
In this paper, the authors have conducted a detailed experimental performance evaluation of Zigbee. The authors have neatly described the contributions of this work, as compared to prior works in Table 2. However, with regards to the performance evaluation, some further clarifications are required. Below are my comments -
1/ In Section 2.1, can't the router nodes also have sensing capability? It's unclear why they are limited to only routing the data and not be equipped with sensors as well.
2/ In line 129, the abbreviation 'Message Authentication Code (MAC)' is the same as 'Medium Access Control (MAC)'
3/ In line 150, what is the relation/mapping between the baud rate and the bit rate?
4/ In section 3, the authors mention about three QoS metrics being considered, while in section 4, there are more than 3 metrics. Please revise for consistency.
5/ In section 4.1.1, doesn't the PDR depend on the bit rate? What bit rate has been considered in the tests?
6/ The figure description/labels are wrong for Figs 6 and 7. The label is the same as Fig 5.
7/ In Figure 9a, is interference from other technologies such as WiFi/Bluetooth being considered? The overall trend for the RSSI descends gracefully/smoothly with distance. However, one would expect more randomness due to the impact of interference. Can the authors consider plotting the (signal to interference plus noise ratio) SINR instead?
8/ In section 4.3, what is the primary cause for such a higher latency when encryption is performed? Is it due to processing delays for AES encryption/decryption? That is surprising.
9/ There are two other aspects that the authors should consider evaluating (at least qualitatively or with some numerical analysis). The below two approaches can help improve the QoS metrics -
i. Data aggregation/compression while routing towards the coordinator. Since data recorded by sensors that are close to each other are correlated, the potential for source coding schemes is great, because a lot of the aggregated data could be compressed before being relayed towards the coordinator.
ii. Channel access: Why not use a TDMA based approach for power saving? What would the synchronization requirements be like, and what kind of algorithm could achieve that?
10/ Lastly, the authors could take a look at the use of Zigbee in a real-world application of seismic acquisition. Below are two studies -
i. "Distributed data acquisition unit based on GPS and ZigBee for electromagnetic exploration," 2010 IEEE Instrumentation & Measurement Technology Conference Proceedings, 2010, pp. 981-985.
ii. "A New Cable-Less Seismograph with Functions of Real-Time Data Transmitting and High-Precision Differential Self-Positioning." Sensors 20, no. 14 (2020): 4015.
Author Response

(The authors gave the same response as above.)

Reviewer 3 Report
This manuscript analyzes extensive experimental results of Xbee S2C modules with various performance metrics such as PDR, power consumption, network life, link quality, latency, and throughput in different environments like indoor vs. outdoor and encrypted vs. unencrypted varying baud rates, transmission powers, hop counts, and transmission distances.
Instead of proposing a new idea, this research is mainly focus on conducting comprehensive performance analyses with as many parameters as possible. Although some experimental results are quite straightforward and easy to expect, there are some interesting analyses as well. Another meaningful result is to find an optimal transmission power level of 3 dBm.
There are some presentation errors as follows.
- In page 11~13, there seem to be Figures and Tables that have wrong numbers.
- (Page 11) The caption of Figure 7 is “(a) indoor and (b) outdoor”, but it seems to be (a) without AES and (b) with AES ?
- (Page 4, line 130) “AES-CTR is mode is used…” -> “AES-CTR mode is used…”
Author Response

(The authors gave the same response as above.)

Reviewer 4 Report
The manuscript deals with performance analysis of the ZigBee network in both indoor and outdoor environments.
The novelty of the paper is very low, as the authors only performed some basic tests on the ZigBee network and presented results achieved for packet delivery ratio, RSSI, delay and energy consumption in the network.
The authors did not propose any novel solutions, some parameters of the experimental setup are not clear, therefore it is hard to replicate nor analyse the presented results.
Authors analyzed link quality using RSSI measurements, this may not be a sufficient metric as link quality is affected mainly by signal to noise and interference ratios. If there is a lot of noise and interference in the channel RSSI value well above the sensitivity of the receiver may not ensure the correct data reception.
In table 1 physical layer of ZigBee standard defines Nominal Tx Power between -32dBm and 0 dBm, however, in experiments the power is between 1dBm and 5dBm.
"Zigbee standard allows for communication in four different topologies: (i) star (ii) tree and (iii) Zigbee Mesh topology [13] as in Figure 2."
What exactly is the fourth topology?
Both "Message Authentication Code (MAC)" and "medium access control (MAC)" use the same abbreviation. Which is a bit confusing for the readers.
"It allows the transmit power up to 8 dBm in boost mode, which can be tuned to three different transmit powers of 1 dBm, 3 dBm, and 5 dBm."
This sentence does not make a lot of sense to me.
"For further illustration, the multi-hop measurement setup for any distance (40 m is taken as an example) is depicted in Figure 3."
Was multihop configuration assumed also for smaller distances?
What does "one or more router nodes" mean? Were router nodes placed in even distances? What was the configuration for example for a distance of 5m between the coordinator and the end node?
It is not clear why there was not a single setup in an indoor environment with LoS propagation. Why authors did not perform the performance test in mixed (LoS and NLOS) conditions?
"Thus, it performs considerably energy efficiently with Zigbee protocol in comparison with the other contemporary technologies like Z-wave, Bluetooth Low Energy (BLE), and LoRa."
Can authors support this claim with some data? Comparison with LoRa might be tricky as LoRa can provide much larger coverage and thus be more efficient in case of long-distance communication.
The dependency of RSSI on distance presented in Figure 9(a) is affected by obstacles, therefore this figure does not provide full information. Results may be different for different positions.
"It is to be noted that the RSSI performance of both the indoor and outdoor environment is good enough for reliable communication as the receiver sensitivity of the Xbee S2C module is -100 dBm."
RSSI alone is not good enough to conclude that the transmission will be successful, it depends on noise and interference as well.
"In an indoor environment, the RSSI improves by 14.73% at 40m which is 10.90% at 5 m of transmission distance while we switch to two-hop communication from a single hop."
This depends significantly on the placement of the "router node".
"Thus, in such a scenario where LoS is minimum with much interference, multi-hop communication is preferred."
This sentence is not clear, what do authors refer to as "much interference"?
From the manuscript is it not clear what was the exact experimental setup. It is not clear how router nodes were placed in the area. It is not clear what routing protocols were used in the network in the case of multi-hop communication. Different routing protocols may have an impact on energy consumption as well as on delay in the network. Therefore such information is important and must be provided in the description of the experimental setup. Moreover, it is not clear if multiple nodes were trying to send data simultaneously, this may have a further impact on the achieved results.
Author Response

(The authors gave the same response as above.)

Round 2
Reviewer 2 Report
The paper has been revised comprehensively and is much improved in terms of clarity and technical details.
Author Response
Thank you for reviewing the manuscript and providing insights to improve quality of the manuscript
Reviewer 4 Report
The setup of the multihop measurement scenario is still unclear. From the results presented in figure 5 a) it is obvious that with transmit power of 5dBm, 100% of packets were delivered when the distance between nodes was less than 35m. Moreover, the only significant drop in packet delivery ratio was in the case when transmit power was set to 1dBm and the distance was 40m.
From these results it is clear that nodes were in communication range most of the time, why would AODV force multi-hop communication in the case when the distance between end node and coordinator was 10 or 20m?
The situation in the outdoor environment was even better, therefore AODV protocol should not force multi-hop communication between nodes since they can communicate directly. What was the setup of devices in this scenario?
"Moreover, Time Division Multiple Access (TDMA) based approach is has shown a great potential in achieving better performances in terms of throughput and energy consumption."
What are the authors trying to say here? ZigBee standard is using probabilistic channel access technique CSMA-CA, doesn't it? Was the TDMA implemented in the ZigBee? Can off-the-shelf devices connect to such a ZigBee network?
The description of figure 9 is not readable since it somehow overlaps with the text.
